# Implication of Different Tumor Biomarkers in Drug Resistance and Invasiveness in Primary and Metastatic Colorectal Cancer Cell Lines

**DOI:** 10.3390/biomedicines10051083

**Published:** 2022-05-06

**Authors:** Marta Sánchez-Díez, Nicolás Alegría-Aravena, Marta López-Montes, Josefa Quiroz-Troncoso, Raquel González-Martos, Adrián Menéndez-Rey, José Luis Sánchez-Sánchez, Juan Manuel Pastor, Carmen Ramírez-Castillejo

**Affiliations:** 1CTB (CTB-UPM) Centro de Tecnología Biomédica, Universidad Politécnica de Madrid, 28223 Pozuelo de Alarcón, Spain; nicolas.alegria.aravena@gmail.com (N.A.-A.); marta.lopez.montes@alumnos.upm.es (M.L.-M.); josefa.quiiroz@gmail.com (J.Q.-T.); raquel.gonzalez@ctb.upm.es (R.G.-M.); menendez.rey.adrian@gmail.com (A.M.-R.); 2Grupo de Sistemas Complejos, Universidad Politécnica de Madrid, 28040 Madrid, Spain; juanmanuel.pastor@upm.es; 3Unidad de Oncología, Hospital de Almansa, 02640 Albacete, Spain; jlsanchez_sanchez@hotmail.com; 4ETSIAAB, Departamento Biotecnología-Biología Vegetal, Universidad Politécnica de Madrid, IdISSC, 28040 Madrid, Spain

**Keywords:** epithelium–mesenchymal transition, cell plasticity, cancer stem cells, tumor biomarkers, resistance, chemotherapy, oxaliplatin

## Abstract

Protein expression profiles are directly related to the different properties of cells and are conditioned by the cellular niche. As an example, they are the cause of the characteristic cell plasticity, epithelium–mesenchymal transition (EMT), and drug resistance of cancer cells. This article characterizes ten biomarkers related to these features in three human colorectal cancer cell lines: SW-480, SW-620, and DLD-1, evaluated by flow cytometry; and in turn, resistance to oxaliplatin is studied through dose–response trials. The main biomarkers present in the three studied lines correspond to EpCAM, CD-133, and AC-133, with the latter two in low proportions in the DLD-1 line. The biomarker CD166 is present in greater amounts in SW-620 and DLD-1 compared to SW-480. Finally, DLD-1 shows high values of Trop2, which may explain the aggressiveness and resistance of these cells to oxaliplatin treatments, as EpCAM is also highly expressed. Exposure to oxaliplatin slows cell growth but also helps generate resistance to the treatment. In conclusion, the response of the cell lines is variable, due to their genetic variability, which will condition protein expression and cell growth. Further analyses in this area will provide important information for better understanding of patients’ cellular response and how to prevent resistance.

## 1. Introduction

Colorectal cancer (CRC) is the third most frequent malignant tumor worldwide (1.8 million cases per year). Moreover, it is the cancer with the third highest mortality rate, representing 8.9% of deaths due to cancer [1]. Five-year survival rates depend on the stage of the tumor, being around 90% for stage I tumors and around 30 60% for stage III tumors [2,3]. Forty percent of cases present as more treatable localized disease, but relapse, which depends on the grade of the primary tumor detected, complicates evolution of the illness [4].

Despite increased incidence in recent years, mortality has decreased, probably due to early detection strategies [5]. The current treatments for CRC consist of surgery, chemotherapy, radiotherapy, and biological drugs. Although biological drug development is evolving, chemotherapy still plays a crucial role when treating colorectal cancer patients. The most common chemotherapy drugs used nowadays are 5-fluorouracil (5-FU), oxaliplatin, irinotecan and capecitabine, with a combination of these usually employed for treatment of advanced CRC [6,7,8].

Oxaliplatin is a platinum compound that binds to plasma proteins and distributes throughout all body tissues, inhibiting the synthesis of DNA in the cells due to crosslinks at susceptible sites in the DNA [9]. The use of oxaliplatin is recommended in advanced disease and metastatic patients, and is usually combined with different drugs [10]. The recommended dosage is 85 mg/m^2^, and it is administered over a period of 2 h, repeated at two-week intervals. If neurotoxicity is persistent, the dose is reduced to 65 mg/m^2^. Moreover, oxaliplatin should always be administered before fluoropyrimidines [11]. Irinotecan is the second line of treatment in patients with oxaliplatin resistance or early disease progression. Its mechanism of action is focused on specifically inhibiting DNA topoisomerase I, inducing lesions in DNA replication and the mitosis process [6,12]. The resistance of patients to treatments is the most worrying cause of low survival rates. Van Der Jeught et. al. [13] describe the different mechanisms of resistance, with the nucleotide excision repair (NER) pathway being the most common mechanism of oxaliplatin resistance.

Although many patients are free of disease after treatment of the primary tumor, around 50% relapse within the first five years after tumor extraction [14]. This is due to the presence of cancer stem cells (CSC), characterized by their resistance to conventional treatments, their auto-renovation capacity, and their slow and asymmetric division [15,16]. During tumor progression, some cancer cells acquire stem cell-like properties and can undergo the epithelial to mesenchymal transition (EMT). These cells invade the extracellular matrix, migrate, and colonize other tissues. A variety of biomarkers have been described to detect CSCs and aid in early diagnosis of relapses. However, none are exclusive to these cells, and there is controversy about their specific role in CRC.

Some biomarkers that were first described for this purpose are BCRP1 (*ABCG2*), CD133 (*PROM1*), CD34 and EpCAM [17,18,19]. In recent years, numerous studies have described novel biomarkers such as LGR5, Trop2 (*TACSTD2*), AC133, CD36, CD166 (*ALCAM*), variants of CD44 (CD44v6), etc. [20,21,22,23,24,25]. Briefly, BCRP1 is a protein transporter able to expel chemotherapy from cells and related to drug resistance [26,27], whereas CD133 is a protein involved in self-renewal whose role (and that of one of its glycosylated forms, recognized by the AC133 antibody) is not yet clear [28,29]. CD34 confers stemness capacity, although it is not only expressed in tumoral cells [18,30,31]. For their part, novel biomarkers such as CD166, EPCAM, LGR5, Trop2, CD44v6 and CD36 are directly involved in adhesion and/or invasion [32,33,34] and are of import in the EMT, thus helping to predict metastasis. The presence of these markers or transient combinations of their expression in small populations with metastatic and EMT capacity may be at the root of the origin of metastases and relapses in patients. These populations, with elusive markers, also called tumor initiating cells or CSCs, are possibly one of the pillars of tumor resistance to treatment, as we see in trials with our model lines.

In this study, we employed three human colorectal cancer cell lines with different metastatic potential: SW-480, SW-620 and DLD-1. The SW-480 cell line is an epithelial cell line from the primary tumor of a human colon adenocarcinoma. The SW-620 cell line was isolated from a metastatic carcinoma from the same patient. Finally, DLD-1 is a cell line that proceeds from a human colon adenocarcinoma in Duke’s C stage. The three cell lines are characterized by the expression of the carcinogenic antigen CEA and p53 mutations. However, positive expression of some oncogenes such as KRAS, myc, myb, fos and sis has been detected in the DLD-1 and SW-620 cell lines. Conversely, SW-480 is the only one that conserves the normal expression of nuclear beta-catenin and the sensitivity to hormonal treatment that proceeds from a human colon adenocarcinoma in Duke´s C stage.

The aim of the present study was to characterize the biomarker expression of these human colorectal cancer cell lines and analyze their resistance to oxaliplatin and irinotecan when exposed to low concentrations over short periods of time.

## 2. Materials and Methods

### 2.1. Cell Culture

Three human colorectal tumor cell lines were employed: SW-480 (CCL-228), SW-620 (CCL-227) and DLD-1 (CCL-221). All of them were purchased from American Type Culture Collection (ATCC, Manassas, VA, USA). The cells were cultured in Dulbecco´s modified Eagle´s medium (DMEM-High Glucose, Dominique Dutscher, Bernolsheim, France) with 10% fetal bovine serum (FBS, PAN Biotech, Aidenbach, Germany), 2 mM L-glutamine (Cytiva, Washington, DC, USA) and 1% penicillin/streptomycin (Corning, New York, NY, USA) at 37 C in a humidified incubator (Series II water Jacker, Thermo Scientific, Waltham, MA, USA) with 5% CO_2_.

### 2.2. Flow Cytometry Analysis

The expression of the different tumor biomarkers was analyzed by flow cytometry. A minimum of three replicates per biomarker were performed. The cells were grown in p100 (Deltalab S.L, Barcelona, Spain) for several days until exponential growth. Then, cells were digested with trypsin and washed twice with PBS 1X (Corning, New York, NY, USA). The pellet was resuspended in PBS + BSA 0.1% (Sigma-Aldrich, St. Louis, MO, USA) and incubated with the corresponding antibody for 15 min at room temperature in darkness. Next, the antibody was removed by centrifugation, and cells were finally detected in a FACS Canto cytometer (Becton Dickinson, Franklin Lakes, NJ, USA).

Anti-BCRP1, anti-CD133, anti-AC133, anti-EPCAM, anti-CD34, anti-CD36, anti-CD44V6 and anti-TROP2 antibodies were purchased from Miltenyi Company (Bergisch Gladbach, Germany) and anti-LGR5 and anti-CD166 antibodies were purchased from Becton Dickinson (Franklin Lakes, NJ, USA). The dilution for each antibody is as described in the following Table 1.

### 2.3. RNA Extraction and Reverse Transcriptase Assay

Cells were harvested with trypsin and centrifuged. Total RNA from cultured cells was extracted using the RNeasy Mini kit (Qiagen, Hilden, Germany) and used immediately for reverse transcriptase reactions or stored at −80 °C until use. RNA concentration was measured in a Nanodrop spectrophotometer (ThermoFisher, Waltham, MA, USA). cDNA synthesis was performed using the RevertAid First Strand cDNA Synthesis kit (Fermentas, Waltham, MA, USA). The reaction was prepared according to the manufacturer’s instructions: a mixture containing 1 µg of RNA, 1 µL of random hexamers and DEPC-H_2_O until 12 µL was incubated at 70 °C for 5 min. Then, 4 µL of buffer 5X, 1µL ribonuclease inhibitor, 2 µL 10 mM dNTP and 1 µL reverse transcriptase were added to each tube. The reaction mixture was then incubated at 25 °C for 5 min, 42 °C for 1 h and inactivated at 70 °C for 10 min. All cDNAs were stored at −20 °C until further use.

### 2.4. Gene Expression by qPCR

The quantitative PCR reaction was performed for *CD133*, *CD166*, *EpCAM* and *TACSTD2* using the KiCqStart SYBR Green kit (Sigma, St. Louis, MO, USA) according to the manufacturer’s instructions. KiCqStart SYBR Green predesigned primers (Sigma, St. Louis, MO, USA) were employed. All primers were used in a final concentration of 300 nM, and 10 ng of cDNA were used per well, for a total volume of 10 µL. All cDNA samples were measured in triplicate in a 96-well plate covered with adhesive seals in the thermocycler Roche LightCycler 480 (Roche, Basel, Switzerland). Reactions started with 10 min at 95 °C, followed by 45 cycles of 15 s at 95 °C, 1 min at 60 °C and 10 s at 72 °C. The 2^−ΔCT^ method was used for calculating the normalized mRNA expression. Beta-actin was used as a housekeeping gene to normalize samples.

### 2.5. Treatment with Oxaliplatin and Growth Curve

The cells were grown in p100 plates for several days until exponential growth. Then, two different plates were seeded: a control plate containing non-treated cells (DMEM without oxaliplatin (Accord Healthcare S.L.U., Barcelona, Spain)) and a plate with treated cells (DMEM with 0.5 µg/mL oxaliplatin). In the case of the DLD-1 cell line, a third plate was seeded with DMEM + 5 µg/mL oxaliplatin. The treated cells were exposed to this high dose of oxaliplatin (0.5 µg/mL or 5 µg/mL in the third DLD-1 plate) for 2 weeks and then maintained at half concentration (DMEM + oxaliplatin 0.25 µg/mL or 2.5 µg/mL in the case of the third DLD-1 plate). These treated and non-treated cells were then evaluated in the growth curve and viability assays.

To analyze the growth, the cells were counted in a Neubauer chamber (Zuzi, Navarra, Spain) at each passage. The total concentration and accumulated number of cells were calculated and represented in a graph.

### 2.6. Cell Viability Detection of Previously Oxaliplatin-Treated and Non-Treated Cells

A thiazolyl blue tetrazolium bromide (MTT, BioChem, PanreacApplichem, Barcelona, Spain) assay was performed to detect the viability of the previously oxaliplatin-treated and non-treated cells exposed to different concentrations of oxaliplatin (050 µg/mL, 10 serial 1:2 dilutions starting from the maximum doses) and irinotecan (Accord Healthcare S.L.U., Barcelona, Spain) (030 µM, four serial 1:2 dilutions from the maximum doses). Briefly, cells were seeded in 96-well plates (Deltalab S.L, Barcelona, Spain) at a concentration of 200,000 cells/mL in a volume of 100 µL. Next, the chosen chemotherapeutic drug was added to each well. Three replicates per condition were performed in each plate with three independent experiments in total. After 72 h of the treatment, the medium was removed, and 50 µL of 0.5 mg/mL MTT was added. Then, plates were incubated for 4 h at 37 °C, and finally, the medium was removed and resuspended in 100 µL of dimethyl sulfoxide (DMSO, Labkem, Barcelona, Spain) before detection of absorbance in a BIOBASE-EL 10A (Biobase, Shandong, China) spectrophotometer at 546 nm.

### 2.7. Statistical Analysis

A non-paired t test was used to compare two groups. One-way analysis of variance (ANOVA) was used to determine the differences between three groups. Wilcoxon test was used to compare the statistical differences between two paired groups, and the Friedman test to compare three or more paired groups. A *p*-value < 0.05 was employed in all of the tests. GraphPad Prism 8 (GraphPad Software Inc., San Diego, CA, USA) and FlowJo (FlowJo LLC, Ashland, OR, USA) were employed to perform the statistics and process the images. Finally, in order to determine the mathematical formulas connecting the decrease in viability with oxaliplatin concentration, we tested linear and exponential decay fits. The fits were performed in Python (Python Software Foundation, Wilmington, DE, USA) with the statistical module “scipy.stats”. Since an exponential function looks like a straight line on a semilogarithmic plot, we performed a linear regression between the logarithm of the viability and the oxaliplatin concentration. 

## 3. Results

### 3.1. Expression of Tumoral Biomarkers in SW-480, SW-620 and DLD-1

The expression of biomarkers that have previously been related to tumorigenesis was quantified by flow cytometry in three human colorectal cell lines (SW-480, SW-620 and DLD-1). Ten different biomarkers with a wide range of roles in cancer progression were analyzed. The mean of the percentages obtained and the standard error of each one is represented in Figure 1. Regarding the SW-480 cell line, three biomarkers were highly expressed. AC133 was expressed in 90 ± 1% of the cells, CD133 in 85 ± 3%, and EPCAM in 100 ± 0%. A similar result was obtained in the SW-620 cell line, with a total of 91 ± 1% of AC133-positive cells, 85 ± 2% of CD133-positive cells and 100 ± 0% of EPCAM-positive cells. However, significant differences were detected when focusing on the CD166 biomarker, as 83 ± 1% of the SW-620 cells expressed it. In relation to the DLD-1 cell line, there were some differences and similarities with the previous cell lines. Regarding the expression of the AC133 and CD133 proteins, these showed less significant expression in the DLD-1 cell line than in SW-480 or SW-620, with percentages of expression of 9 ± 2% and 20 ± 2%, respectively. The CD166 protein was expressed at levels as high as in the SW-620 cell line, which were significantly different from levels in the SW-480 cell line (80 ± 1% of CD166-positive cells). Moreover, TROP2 protein was significantly highly expressed when compared to SW-480 and SW-620 cell lines (99 ± 0.2% TROP2-positive cells in DLD-1 versus 0% in the rest). Finally, EPCAM was highly expressed in all three cell lines (100 ± 0%).

A representation of the histograms obtained for each biomarker and each cell line is shown in Figure 2. The controls (cells without staining) are represented in black, and the distribution of stained cells for each biomarker is represented in red (SW-480), green (SW-620) or orange (DLD-1). The previously described differences are clearly seen in this figure, as the histograms corresponding to CD133, AC133, CD166, EPCAM and TROP2 biomarkers move to the right side of the graph in the case of expressed biomarkers. The corresponding percentage of positive cells is also shown in each image for this representative replicate.

Finally, the expression of biomarkers that showed differences between cell lines in the previous flow cytometry experiment was confirmed via RT-qPCR assays. Normalized expression is shown in Figure 3, using beta-actin as a reference gene because of its constitutive cell expression. Regarding CD133 (*PROM1*), the SW-480 and SW-620 cell lines presented a higher significant expression when compared to DLD-1. This result corresponds with that obtained in the flow cytometry assay. In the case of CD166 (*ALCAM*), the SW-480 cell line expressed this biomarker the least, showing higher values in the SW-620 and DLD-1 cell lines. A similar profile was obtained by flow cytometry. *EPCAM* expression was also quantified, obtaining high values (note: see the scale) in every cell line. However, the expression of *EPCAM* in DLD-1 was significantly higher than in the other cell lines. Finally, the *TACSTD2* expression profile was similar tothe cytometry assay, with no expression in the SW-480 and SW-620 cell lines and significantly different expression in DLD-1.

### 3.2. Colorectal Cell Line Growth in Exposure to Oxaliplatin Chemotherapy Drug

The growth of colorectal cell lines differed when they were exposed to the oxaliplatin chemotherapy drug. Figure 4a shows how the number of cells of the three human colorectal cell lines fell when comparing non-treated (0 µg/mL oxaliplatin) to treated (0.5 µg/mL oxaliplatin) cell lines. Although in the images it seems that the DLD-1 cell line grew the fastest, the cell shape should be considered. The counting of cells for each cell line and condition (Figure 4a, bottom right) showed that, in fact, SW-620 grew the fastest, followed by SW-480 and DLD-1. Moreover, treatment with oxaliplatin reduced the number of cells in all cases.

Cells were seeded for 2 weeks in exposure to oxaliplatin; the number of cells for each passage is represented in Figure 4b. In all cases, an exponential curve was obtained as expected for cell growth. However, the split rate was higher in the SW-620 cell line, followed by the SW-480 and DLD-1 cell lines. Moreover, we showed that the growth curve was totally flattened when cells are treated with 0.5 µg/mL oxaliplatin in the case of SW-480 and SW-620. A specific behavior was seen in the DLD-1 cell line, as this concentration of oxaliplatin (0.5 µg/mL) reduced the number of live cells but was not enough to flatten the DLD-1 cell line growth curve. For that purpose, a higher dose of oxaliplatin was employed (5 µg/mL oxaliplatin), obtaining results similar to those obtained in the other cell lines with the 0.5 µg/mL dose.

### 3.3. Resistance to Oxaliplatin in Non-Treated and Previously Oxaliplatin-Treated Cell Lines

The resistance to oxaliplatin was measured in the SW-480, SW-620 and DLD-1 cell lines, using previously oxaliplatin-treated and non-treated cells. The resistance to oxaliplatin was evaluated in a wide range of oxaliplatin concentrations (0–50 µg/mL).

Regarding the SW-480 and SW-620 non-treated cell lines (Figure 5), gradual decreases in cell viability were observed at low oxaliplatin doses (0–0.4 µg/mL), and this was maintained until a 3.25 µg/mL oxaliplatin dose, presenting a resistant population of 63 ± 2% and 62 ± 0.5%, respectively. At this point, a slight decrease was observed until a 25 µg/mL dose, reaching a cell viability of 12 ± 0.6% for both lines. Finally, the curve flattened at a resistant population of 11 ± 2% and 10 ± 0.3% for the SW-480 and SW-620 cell lines, respectively.

The DLD-1 non-treated cell line (Figure 5) showed less of a decrease in cell viability at the 0.4 µg/mL oxaliplatin dose, with a viability percentage of 78 ± 7% compared to values of 64 ± 3% and 63 ± 3% in the other evaluated cell lines. The growth curve flattened at a resistant population of 50 ± 3%. Afterwards, cell viability was maintained until the 15 µg/mL oxaliplatin dose, at which point it steadily decreased until the last tested dose (50 µg/mL), obtaining a final cell viability of 11 ± 3%.

In order to determine the mathematical formula relating the decrease in viability to the oxaliplatin concentration, we tested linear and exponential decay fits. Two ranges of oxaliplatin concentrations were evaluated: 0–0.4 µg/mL and 3.25–25 µg/mL in SW-480 and SW-620, and 0–0.4 µg/mL and 12.5–50 µg/mL in DLD-1. These concentrations were used because cell viability was maintained in the intermediate range, as previously described. The exponential fits (Figure 6) reported better Pearson correlation coefficients and lower *p*-values (*p* < 0.05 for all conditions, except for non-treated SW-480), meaning the cell viability decreased exponentially at low and high ranges of oxaliplatin concentrations.

Regarding the previously oxaliplatin-treated cells, a profile similar to that of the control non-treated cells was obtained. Despite this similar profile, there were significant differences between non-treated and previously oxaliplatin-treated cells in all cell lines. The cell viability percentages were significantly higher in previously oxaliplatin-treated cells when compared to control non-treated cells (Figure 7a), which means that they were more resistant to oxaliplatin. In order to discard the possibility of their presenting general resistance, another chemotherapy drug was also evaluated. The results in Figure 7b show that non-treated and previously oxaliplatin-treated cells presented no significant differences in relation to the response to irinotecan. A gradual decrease in cell viability was observed in every cell line when exposed to irinotecan, and resistant populations of 55 ± 3% in the SW-480 cell line, 56 ± 6% in SW-620 and 66 ± 8% in the DLD-1 cell line were shown at the maximum irinotecan dose evaluated (30 µM).

All of these results led us to conclude that previously oxaliplatin-treated cells were resistant to the same drug to which they were exposed (oxaliplatin) but not to others (irinotecan).

### 3.4. TROP2 Role in Drug Resistance

We have already shown that TROP2 is the biomarker with high expression in DLD1, with this line being the most resistant to oxaliplatin. Moreover, we have demonstrated that cells previously exposed to oxaliplatin are more resistant than non-exposed cells. In this section, we wanted to test whether the expression of *TACSTD2* increased or not in previously exposed cells to evaluate the role of TROP2 in the resistance to oxaliplatin. Figure 8 shows the normalized expression of *TACSTD2* versus beta-actin (reference gene). The SW-480 and SW-620 cell lines did not present *TACSTD2* expression in non-treated or treated cells. However, significant differences were found between previously oxaliplatin-treated (at both doses: 0.5 and 5 µg/mL oxaliplatin) and non-treated DLD1 cells.

## 4. Discussion

The present work aimed to characterize tumor biomarker expression (diagnostic tool) and the response to chemotherapy (resistance to drugs) of human colorectal cell lines. We selected three different colorectal cell lines: the first from a primary tumor, the second from a metastatic tumor, and the third characterized by its resistance to chemotherapeutic drugs. We showed the differential distribution of a variety of controversial biomarkers’ expression in three representative cell lines with different tumor profiles. Ten different biomarkers related to metastasis, invasion, and adhesion were chosen. All of them had been previously described in the literature as cancer stem cell biomarkers, although their specific roles are still under debate [35]. Although some studies have been performed on the individual expression of these biomarkers in different tissues or cell lines, obtaining different and non-consistent results [19,36], in this study we analyzed a total of ten of them using the same method across the board and comparing their expression in three different tumor profiles. Due to the function of the biomarkers that showed differential expression the tested cell lines, all of these differences were based on the cancer cells’ EMT ability, as described in further detail below.

In our experiments we showed that the SW-480 and SW-620 cell lines presented a similar expression profile, but CD166 was overexpressed in SW-620. SW-480 proceeded from a primary adenocarcinoma cancer, while SW-620 came from the metastasis in the same patient. This indicates that CD166 may play an important role in metastasis. This was confirmed by studies that support that this biomarker is capable of promoting metastasis by interacting with other molecules, such as SOSTDC1 [37,38]. Moreover, the DLD-1 cell line also expressed this protein at high levels. CD166 is a transmembrane protein that belongs to the immunoglobulin superfamily of cell adhesion molecules (Ig-CAMs), which mediates intercellular adhesion [39], essential for the EMT. A recent study showed that CD166 appears to be an EMT epithelial phenotypic molecule in response to TWIST-induced EMT in colorectal cancer cells, and the response seems to be dependent on the microsatellite instability of the cells [40]. The DLD-1 cell line is described as microsatellite instable (MSI), whereas SW-480 and SW-620 are microsatellite stable (MSS), which could explain the high expression of CD166 in DLD-1, as shown in Figure 1, Figure 2 and Figure 3.

Another significant difference was found in relation to the Trop2 protein. Trop2 is a transmembrane glycoprotein that transduces its signal by means of calcium upon binding to PIP2 in its intracellular domain. In this study, we evaluated its expression and its relation to drug resistance. We showed a significantly higher expression of TROP2 (or *TACSTD2*) in the DLD-1 cell line when compared to that in the SW-480 and SW-620 cell lines, which were more resistant to chemotherapy. Moreover, we demonstrated that previously oxaliplatin-exposed cell lines are more resistant to oxaliplatin and, in the case of DLD-1, present higher expression of *TACSTD2.* Although further experiments are needed, all of these results together show a correlation between the expression of this protein and a drug-resistant phenotype, a promising result for predicting the response to oxaliplatin in patients.

Some authors have described Trop2 as a predictor of poor patient survival and have related it to the chance of disease recurrence and liver metastasis in colon cancer [22]. However, the relation between Trop2 expression and drug resistance is little studied. Several reports have suggested that Trop2 expression regulates tumor cell resistance to therapeutic drugs, including tamoxifen and trastuzumab, among others [41,42], which could be explained by its effect regulating the Notch1 signaling pathway in some cells. Similarly, Guerra et al., 2016 show how the expression of Trop-2 generates cell survival by the activation of AKT [43,44]. Moreover, Sun et al., 2021 state that Trop-2 allows for remodeling of the tumor microenvironment, leading to resistance to different drugs [45]. In conclusion, although further molecular experiments are needed to confirm the relation between Trop2 and drug resistance, our work together with those mentioned are an important contribution to the description of the new role for the Trop2 protein in drug resistance.

The role of Trop2 should be considered together with EpCAM since both proteins share certain functions and present 50% sequence identity. In fact, Trop2 might be a modulator and/or enhancer of EpCAM-induced cell signaling [46]. The roles of EpCAM include cell–cell adhesion, proliferation, migration, invasion, and differentiation. This protein is overexpressed in 80–100% of colorectal cancers [47] and is, in fact, used for several diagnostic and therapeutic tools, such as the anti-EpCAM antibody described by Liao et al. [48] for colon cancer treatment. In this study, EpCAM was highly expressed in every cell line, as we showed both in flow cytometry and RT-qPCR assays. Moreover, higher gene expression was found in the case of the DLD-1 cell line, which correlated with the high expression of Trop2 in DLD-1. In fact, EpCAM has also been previously related to drug resistance by some authors [49]. Further studies could be performed regarding the interaction of these two proteins, EpCAM and Trop2 in DLD-1, as this cell line might serve as a model cell line for their study.

It is important to consider that there is heterogenicity in the expression profiles of different cell lines, as occurs in patients. This is why some authors declare that elucidating the differences in Trop2 expression in certain cancers and disease stages would be vital to uncover its exact role in cancer growth and metastasis [41]. This is what we aimed to contribute with this work, not only for Trop2 but the remaining different biomarkers.

As regards the CD133 protein, there are different opinions as regards its function. This protein was first described to detect cancer stem cells, but in recent years, its role in predicting prognosis has been of more interest. Some studies have related the expression of this protein with the cell cycle stage of the cells [50], meaning that its expression is very variable. CD133 and its epitope, AC133, active molecules in cell self-renewal and quiescence, may be involved in the recent hypothesis regarding the importance of dormancy and quiescent cells in the resistance and progression of neoplastic disease [50,51,52].

We have studied the expression of the CD133 and AC133 clone, the glycosylated version of the protein. In both cases, expression was high in SW-480 and SW-620 and decreased in DLD-1, which suggests that SW-480 and SW-620 cells are potentially less differentiated cells with self-renewal potential. Although high levels of CD133 have been related to drug resistance in other studies [53,54], we were not able to establish this correlation in these cell lines.

An important issue is that the biomarkers that have been described to date are not individually exclusive to CSCs. The combined study of a variety of them would be necessary to gain valuable information about their role in cancer progression.

Since these are previously described biomarkers, certain levels of expression are expected, but these vary in certain conditions and between patients and cell lines. One example is the BCRP1 protein, for which we did not obtain significant expression data. Huang et al. [55] describe that this protein is weakly expressed in SW-480, as we have also shown, but is highly expressed in the HT-29 cell line. The same occurs for other non-expressed biomarkers. Another example could be CD36, a lipid transporter. It has been previously described that lipid metabolism is not representative [56] in certain cell lines such as SW-480 and SW-620, so it makes sense that the expression of this transporter is low. Finally, levels of CD44V6 have been described to be low in the SW480 cell line and overexpressed when exposed to 5FU [57]. We were able to confirm these findings and to compare this cell line with two others, obtaining similar results.

Biomarker characterization is important in terms of understanding the different proteins that are expressed in the tumor microenvironment compared to normal samples. This is also important to describe new diagnostic methods for early detection of metastasis, and the reason why basic information on the expression in different cell lines and patients is valuable. Furthermore, the evaluation of the pharmacological response of the cells to drugs and the growth rates of different cell lines is essential for better determination of what is occurring in patients that are resistant to some treatments. In fact, the morphological response of cells helps to predict cancer patients’ prognosis [58], where some malignant cells are capable of increasing their size to contain vacuolar drugs [59].

Here, we demonstrated that the growth of the different tumor cell lines varies, as seen in Figure 3. SW-620 was the fastest growing cell line, and this could be explained by the active molecular pathways and its metastatic origin. Moreover, we treated the three cell lines with the most common chemotherapeutic drug administered to patients as a first line of treatment, oxaliplatin. We could see how the growth of the SW-480 and SW-620 cell lines drastically decreased, obtaining a flattened growth curve at 0.5 µg/mL oxaliplatin. This effect was not shown in the DLD-1 cell line, being more resistant to oxaliplatin, as the number of cells decreased, but the curve was still exponential. A higher dose of oxaliplatin was provided in this case to obtain the same effect and study the resistance to chemotherapy. Some other studies have demonstrated the resistance over longer periods of time [60,61,62]. The purpose of this study was to demonstrate that exposure to oxaliplatin in a low dose over short periods of time can lead to resistance to the drug.

We analyzed the growth curve for each cell line, with SW-480 and SW-620 presenting similar profiles. A first exponential decrease in cell viability until 0.4 µg/mL oxaliplatin concentration was detected, followed by a lineal maintenance and a slight exponential decrease from 3.25 µg/mL oxaliplatin concentration. In the DLD-1 cell line, a less-pronounced, although also exponential, decrease was observed, followed by a greater flattened period from 0.4 to 12.5 µg/mL oxaliplatin. Then, an exponential decrease was observed from 12.5 to 50 µg/mL oxaliplatin concentration.

All of these results showed that the SW-480 and SW-620 cell lines presented the same response profile to oxaliplatin, whereas the DLD-1 cell line was more resistant to it, as there were a higher number of viable cells when comparing the same low dose of oxaliplatin, and cell viability was maintained over a larger range of concentrations. Moreover, previously oxaliplatin-treated cells were more resistant in every case, as a lower viability percentage was detected in all of them when compared with non-treated cells of the same cell line. Finally, we demonstrated that the mechanism of resistance is specific to the drug used, as can be seen when comparing to cells exposed to irinotecan, the second treatment option in colorectal cancer patients. This can be explained because the resistance to oxaliplatin or irinotecan presents some differences, as oxaliplatin resistance is related to the nucleotide excision repair (NER) pathway and irinotecan resistance has been related to epigenetic modifications such as histone acetylation [13]. This is an important and highly encouraging finding for cancer patients.

To conclude, we found differences in the expression of biomarkers that allowed us to discern between different tumorigenicity profiles. On the other hand, exposure to oxaliplatin slows cell growth but generates resistance to it. The tumor microenvironment, tumor malignancy and the EMT capacity can condition both protein expression and cell growth in response to resistance. Further analyses in this area will provide important information for improved understanding of the cellular response of different colorectal tumors to chemotherapy, their resistance to treatment and how to prevent it.

## Figures and Tables

**Figure 1 biomedicines-10-01083-f001:**
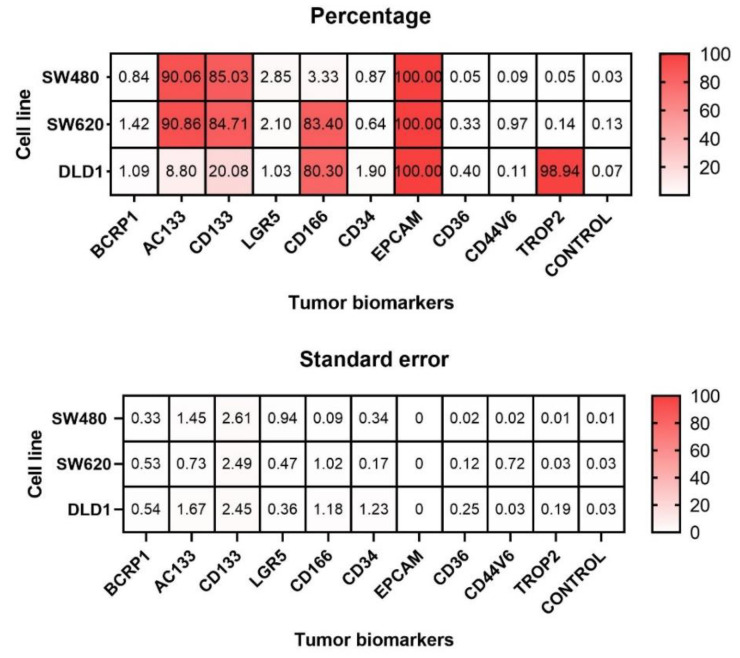
**Expression of tumor biomarkers in three colorectal cell lines (SW-480, SW-620 and DLD-1).** The percentage and standard error are shown for each cell line and tumor biomarker (BCRP1, AC133, CD133, LGR5, CD166, CD34, EPCAM, CD36, CD44V6 and TROP2).

**Figure 2 biomedicines-10-01083-f002:**
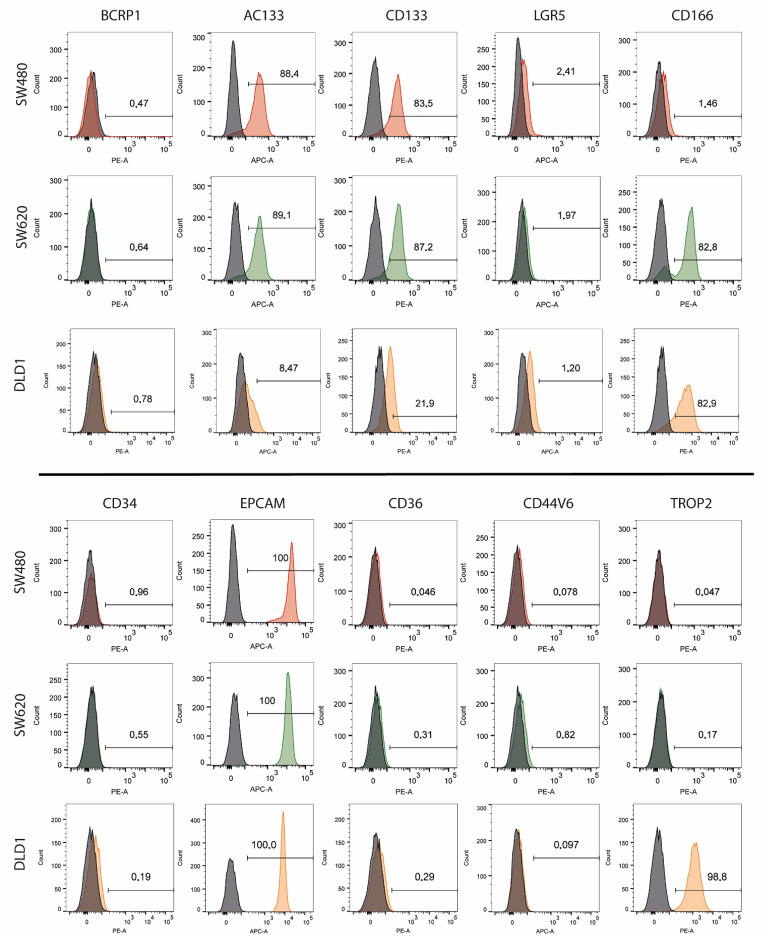
**Representative histogram images of tumor biomarker expression for each cell line.** A representative replicate is shown for each biomarker and cell line. The control is represented in black, and the biomarker in red (SW-480), green (SW-620) or orange (DLD-1). The control is used to determine the gate from which cells are going to be considered positive. The percentage of positive cells is detailed in each image.

**Figure 3 biomedicines-10-01083-f003:**
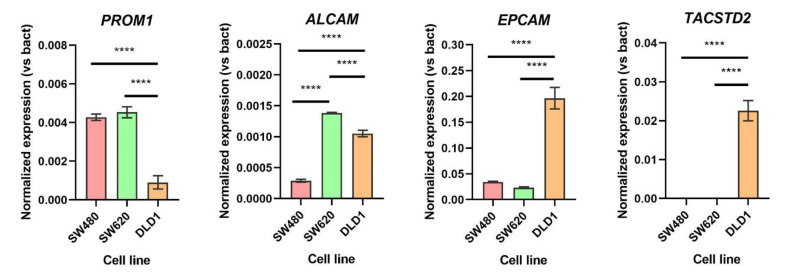
**Expression of tumor biomarkers in three colorectal cell lines (SW-480, SW-620 and DLD-1).** The mRNA expression of *PROM1*, *ALCAM*, *EPCAM* and *TACSTD2* was measured in the SW-480, SW-620 and DLD-1 cell lines. β-actin gene was used to normalize the gene expression, measured in triplicates. Significant differences are indicated (ANOVA test, **** *p* < 0.0001).

**Figure 4 biomedicines-10-01083-f004:**
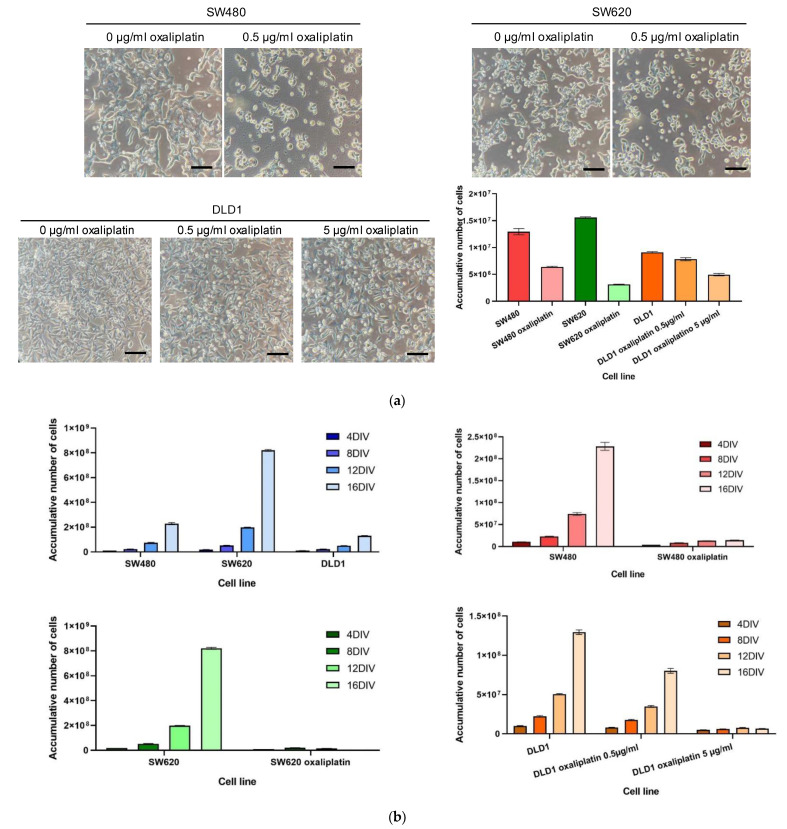
**Growth pattern of non-treated and previously oxaliplatin-treated cells in SW-480, SW-620 and DLD-1 cell lines.** Cells were treated for 2 weeks at a concentration of 0.5 µg/mL oxaliplatin. A concentration of 5 µg/mL was also used in the DLD-1 cell line because of its known resistance to drugs. (**a**) Micrographs and counting of each cell line and condition. Scale bar 100 µm. (**b**) Growth curve comparing the three non-treated cell lines (top-left) and treated versus non-treated cells in SW-480 (top-right), SW-620 (bottom-left) and DLD-1 (bottom-right). Three different measures were performed.

**Figure 5 biomedicines-10-01083-f005:**
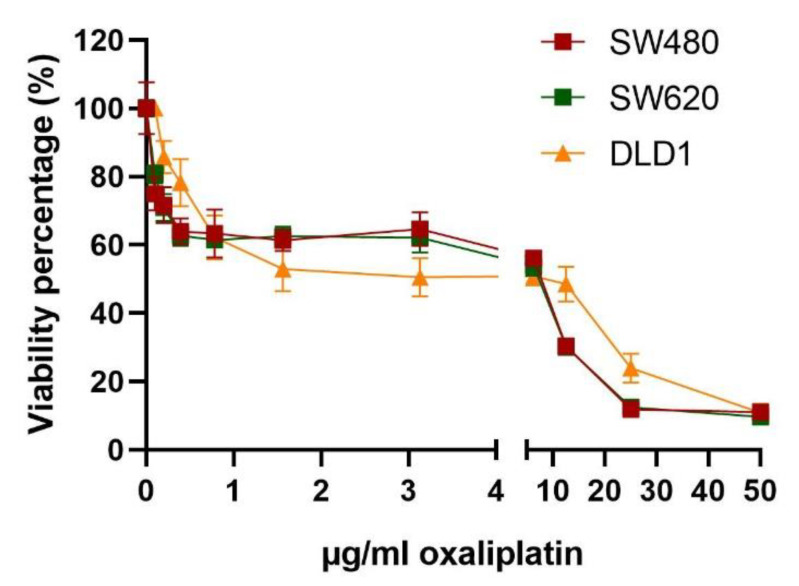
**Resistance of SW-480, SW-620 and DLD-1 cell lines to chemotherapy.** Resistance to oxaliplatin in a range of 0 to 50 µg/mL was evaluated. A decrease at low oxaliplatin concentrations followed by maintenance of cell viability and then another decrease at high oxaliplatin concentrations was shown.

**Figure 6 biomedicines-10-01083-f006:**
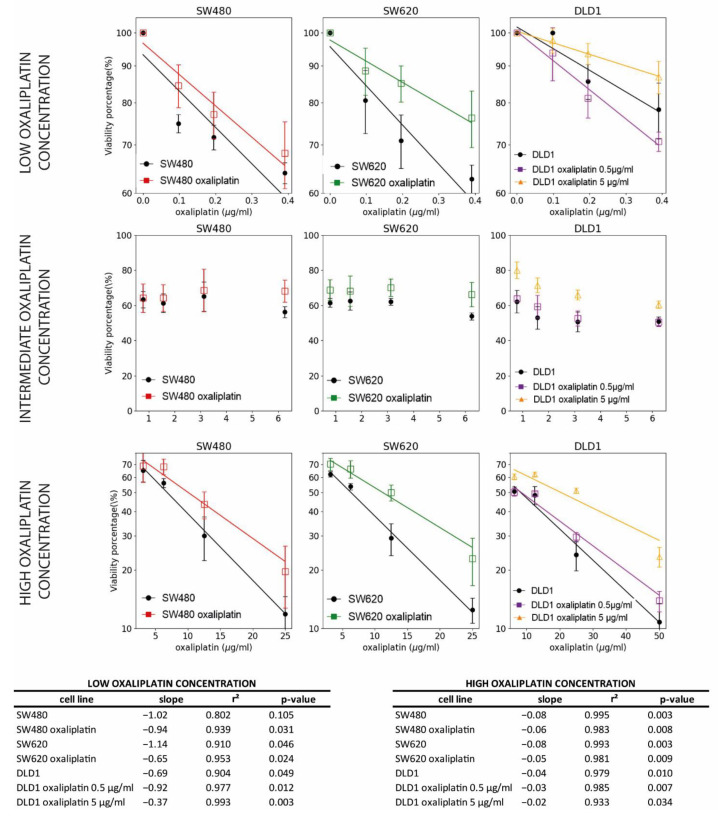
Exponential decrease in cell viability when SW-480, SW-620 and DLD-1 cell lines are exposed to low and high oxaliplatin concentrations. A linear trend in the log-lin plots is a sign of exponential decay. The slope, r2 and *p*-value are shown in the table for each condition. Note that the viability percentages are in log scale and the exponential fits are straight lines.

**Figure 7 biomedicines-10-01083-f007:**
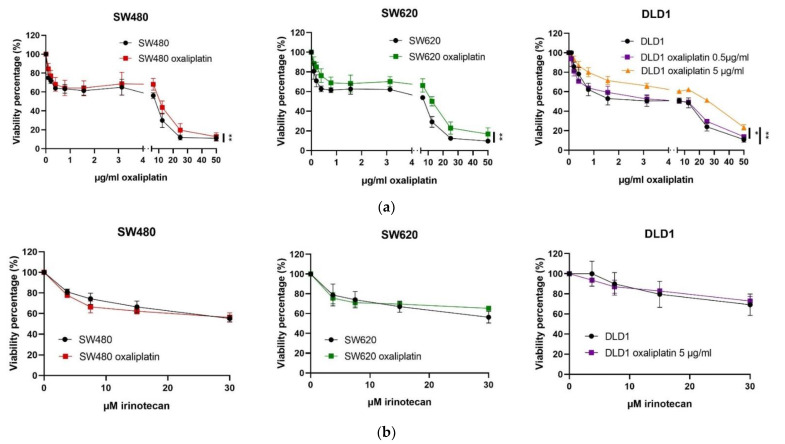
**Resistance of previously oxaliplatin-treated and non-treated SW-480, SW-620 and DLD-1 cells to chemotherapy.** (**a**) Resistance to oxaliplatin in a range of 0 to 50 µg/mL, (**b**) resistance to irinotecan in a range of 0 to 30 µM. Previously oxaliplatin-treated cells were more resistant to oxaliplatin but not to irinotecan in all three cell lines. Significant differences are indicated as ** *p* < 0.01, * *p* < 0.05.

**Figure 8 biomedicines-10-01083-f008:**
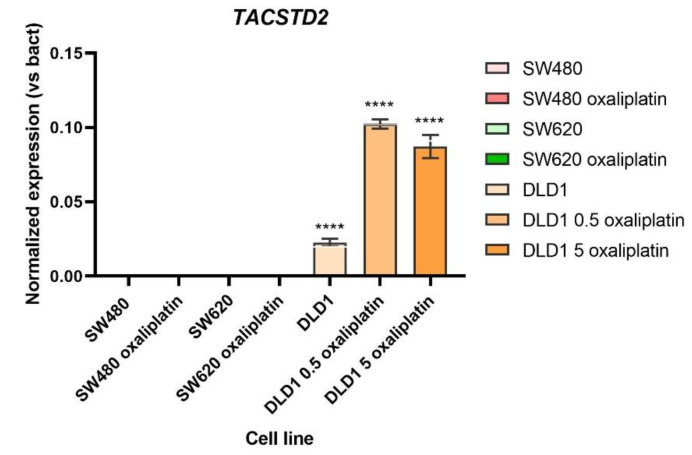
***TACSTD2* expression in previously oxaliplatin-treated and non-treated SW-480, SW-620 and DLD-1 cells.** Previously oxaliplatin treated-cells present significantly higher expression in the DLD-1 cell line when compared to non-treated cells. In SW-480 and SW-620, no significant differences were detected, as there was no expression in all of those cases. Significant differences are indicated as **** *p* < 0.0001.

**Table 1 biomedicines-10-01083-t001:** Antibody dilution for flow cytometry application.

Antibody	Dilution	Antibody	Dilution
anti-BCRP1	1:11	anti-CD36	1:50
anti-CD133	1:50	anti-CD44V6	1:50
anti-AC133	1:50	anti-TROP2	1:100
anti-EPCAM	1:50	anti-LGR5	1:50
anti-CD34	1:50	anti-CD166	1:11

## Data Availability

All data are available in the main text.

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
