# Peer review of "Implication of Different Tumor Biomarkers in Drug Resistance and Invasiveness in Primary and Metastatic Colorectal Cancer Cell Lines"

_biomedicines, 2022, doi:10.3390/biomedicines10051083_

Round 1

Reviewer 1 Report

       The submitted manuscript written by Sánchez-Díez et al. contains two types of experiments: one is an extensive FACS study, detecting the level of ten already described tumour markers expressed in three different colorectal cancer cell lines, and the second experimental setting is a cell culture series, measuring the oxaliplatin-susceptibility of these cell lines.

        According to the abstract the characterization of ten biomarkers will be connected to the emergence of oxaliplatin-resistance. This research objective is clear, but the work presented here did not relate the FACS profile of the cell lines to the drug susceptibility. The main drawback of this work is, that this work is not ready, yet and the cohesion between the two types of experiments is missing.

There is a confusing part in the results and the conclusion, the authors should have discriminated better the potential role of the measured biomarker expression:

1., The high expression level of AC133 and CD133 cancer stem cell marker on the SW cells predicts that the SW cells are potentially less differentiated cells with high self-renewal potential and in this cell lines the high CD133 marker level is not linked to drug resistant phenotype. (Although high level of CD133 could lead to drug resistance in other cells.)

2., Based on the presented results, the most promising candidates for predicting the response to oxaliplatin treatment is the measurement of Trop2 level. (Although Trop2 is a described marker for invasiveness and not for the drug resistance.)

These statements are not included clearly in the manuscript although the results obtained here imply these conclusions.

As the involvement of Trop2 in drug resistance is a new concept, this part of the work should be justified experientially. Therefore, I suggest proofing this notion and perform at least two types of experiments.

1., Downregulate Trop2 level (via siRNA) in DLD1 cells and challenge the oxaliplatin-susceptibility of the Trop2-silenced DLD1 cells.

2., Check the potential increase of Trop2 expression level (via FACS or WB) on the oxaliplatin-resistant cell fractions of the three cell lines.

The most interesting part of the manuscript is that how can be linked the Trop2 (calcium signal transducer membrane protein) expression to the oxaliplatin (which is a DNA modifying agent) resistance. Trop2 is mostly related to metastasis formation of tumour cells and the drug-resistance-promoting effect was not widely studied (found only one paper suggesting that Trop2 is involved in the development of oxaliplatin-resistance /DOI: 10.1159/000491587/). Therefore, the results of the suggested experiments could represent the major novelty of this manuscript.

Minor mistakes:

-Change the title because it is not entirely linked to the content of the manuscript.„Resistance generation in primary and metastatic colorectal cancer”: suggests that patients-derived colorectal cancer will be analysed and not cell lines. Furthermore, „notes for future precision medicine and EMT implications”, this part is missing from the manuscript.

-line 26: Correct the sentence in the Abstract: “In conclusion, the response of the cell lines is variable, due to the tumor microenvironment, which will condition protein expression and cell growth.” Here it is not clear how the tumour microenvironment was implicated in the topic, as the drug response variabilities was measured in artificial tumour microenvironment of the in vitro cell culture.

-line 202.: Please use another word in this sentence: “the cell extension should be considered”. The phrase “cell size” or “cell shape” would be better.

-Figure 3: Include statistics and the error bars on the column and write the number of repetitions in the figure legends.

Line 119: Please, include the dilutions of the used antibodies instead of referring to the datasheet description.

-line 278: instead of “We show the differential contribution of a variety of controversial biomarkers expression in three representative cell lines with different tumor profiles.” Better to write: „We show the differential distribution of a variety of controversial biomarkers...”

-line 318: the sentence is not correct: “Trop2 modulation of EpCAM can cause EpCAM to proliferate and migrate into liver parenchyma.”

 -Reference 48 is not correctly cited.

Author Response

Contact Name: Carmen Ramírez Castillejo
Title: Doctor in Cell Biology

Address: Cancer Stem Cell laboratory, HST Research Group.

Centro de Tecnología Biomédica

Dpto. of Biotechnology-VB. ETSIAAB.

Universidad Politécnica de Madrid (UPM)

Avda/ Puerta de Hierro 2-4. 28040 Madrid. Spain

Dear Reviewer,

First of all, I would like to thank you for your time and effort reviewing the manuscript. All of your comments were of great interest, allowing us to improve different aspects of the research article. In this letter we detail the main changes to the article in answer to your comments. We have tried to incorporate as many changes as possible attending to your suggestions, and we hope it is now more suitable for publication in this Special Issue from Biomedicines.

POINT BY POINT CHANGES:

There is a confusing part in the results and the conclusion, the authors should have discriminated better the potential role of the measured biomarker expression:

We agree the explanation of the results does not adequately connect different sections. We have therefore modified some aspects in the discussion, to provide that connection.

We have selected three cell lines: SW-480, SW-620 and DLD-1. Each cell line has a different origin, with the first coming from a primary tumor, the second a metastatic tumor, and the third from a primary tumor that is characterized by a drug resistant phenotype. This selection allows us to discuss invasiveness (when comparing biomarker expression for the two first cell lines) and drug resistance (when comparing biomarker expression of the first two with the third, or the same cell line between previously oxaliplatin exposed cells and non-treated cells).

1., The high expression level of AC133 and CD133 cancer stem cell marker on the SW cells predicts that the SW cells are potentially less differentiated cells with high self-renewal potential and in this cell lines the high CD133 marker level is not linked to drug resistant phenotype. (Although high level of CD133 could lead to drug resistance in other cells.)

You are right, we were only describing the variability in the expression but did not go into depth. We have incorporated your suggestion in lines 436-441 (numeration with Track changes turned on), where we present the potential self-renewal of SW cells and reference some studies where the expression of this biomarker is high and related to drug resistance in other cell lines (although we were not able to establish this correlation in the selected cell lines).

2., Based on the presented results, the most promising candidates for predicting the response to oxaliplatin treatment is the measurement of Trop2 level. (Although Trop2 is a described marker for invasiveness and not for the drug resistance.)

These statements are not included clearly in the manuscript although the results obtained here imply these conclusions.

As the involvement of Trop2 in drug resistance is a new concept, this part of the work should be justified experientially. Therefore, I suggest proofing this notion and perform at least two types of experiments.

1., Downregulate Trop2 level (via siRNA) in DLD1 cells and challenge the oxaliplatin-susceptibility of the Trop2-silenced DLD1 cells.

2., Check the potential increase of Trop2 expression level (via FACS or WB) on the oxaliplatin-resistant cell fractions of the three cell lines.

The most interesting part of the manuscript is that how can be linked the Trop2 (calcium signal transducer membrane protein) expression to the oxaliplatin (which is a DNA modifying agent) resistance. Trop2 is mostly related to metastasis formation of tumour cells and the drug-resistance-promoting effect was not widely studied (found only one paper suggesting that Trop2 is involved in the development of oxaliplatin-resistance /DOI: 10.1159/000491587/). Therefore, the results of the suggested experiments could represent the major novelty of this manuscript.

This comment was really useful for us as we agree that the original version did not deepen on this important issue. In light of this we have changed the discussion from lines 380-406 (numeration with Track changes) to better answer this point. We present the results obtained for this biomarker, new references related to Trop2 and drug resistance and performed some of the experiments you suggested in your revision.

Although some authors have worked on this, the link between Trop2 and oxaliplatin is little studied as you have mentioned. Nevertheless, we have tried to justify this both experimentally and with bibliographical references. Unfortunately, we were not able to execute the downregulation experiment you suggested in the 10 day revision period, although we think it is quite interesting and hope to be able to explore this idea further in the coming months. However, we have studied the expression of TACSTD2 (gene that encodes TROP2 protein) by RT-qPCR. We have incorporated these results in a new figure (Figure 8). As expected, DLD1 previously oxaliplatin exposed cells present higher levels of TACSTD2, confirming the role of Trop2 in drug resistance. Although further experiments are needed in this area, we think this is sufficient to explain the differential biomarker expression between cells and its relation to drug resistance.

Minor mistakes:

-Change the title because it is not entirely linked to the content of the manuscript.„Resistance generation in primary and metastatic colorectal cancer”: suggests that patients-derived colorectal cancer will be analysed and not cell lines. Furthermore, „notes for future precision medicine and EMT implications”, this part is missing from the manuscript.

We have changed the title to something more specific to the content of the article: “Implication of different tumor biomarkers in drug resistance and invasiveness in primary and metastatic colorectal cancer cell lines.”

-line 26: Correct the sentence in the Abstract: “In conclusion, the response of the cell lines is variable, due to the tumor microenvironment, which will condition protein expression and cell growth.” Here it is not clear how the tumour microenvironment was implicated in the topic, as the drug response variabilities was measured in artificial tumour microenvironment of the in vitro cell culture.

We totally agree with your comment, we have replaced the term “tumour microenvironment” with “genetic variability”, as the experiment is performed in different cell lines and this more aptly describes what can be modifying the expression and/or regulation of the different proteins.

-line 202.: Please use another word in this sentence: “the cell extension should be considered”. The phrase “cell size” or “cell shape” would be better.

We have replaced it to cell shape, as this more aptly describes what we wanted to say.

-Figure 3: Include statistics and the error bars on the column and write the number of repetitions in the figure legends.

We have included your suggestions.

Line 119: Please, include the dilutions of the used antibodies instead of referring to the datasheet description.

We have included this information in a new table.

-line 278: instead of “We show the differential contribution of a variety of controversial biomarkers expression in three representative cell lines with different tumor profiles.” Better to write: „We show the differential distribution of a variety of controversial biomarkers...”

It has been replaced in the corresponding line.

-line 318: the sentence is not correct: “Trop2 modulation of EpCAM can cause EpCAM to proliferate and migrate into liver parenchyma.”

We have re-considered this sentence and have decided to remove it from the text.

 -Reference 48 is not correctly cited.

Thank you, it has been corrected.

Finally, we wanted to mention that a new figure has been added (Figure 3) to confirm the results obtained by flow cytometry. This was suggested by the other reviewer and we considered it interesting for the article. The article’s English has also been revised (please see the corresponding Certificate of Proofreading) as was suggested.

We hope these corrections are satisfactory for you and look forward to your response. Thank you for your time.

Best Regards,

Carmen Ramírez Castillejo

Reviewer 2 Report

The manuscript by Sánchez-Díez et al. assesses the relation between the expression of ten cancer stem cell biomarkers and drug resistance to oxaliplatin in three colorectal cancer cell lines. The choice of model cancer cell lines is reasonable in that they have different metastatic potential and the expression of the biomarkers is sufficiently diverse to allow for relevant conclusions, yet the methods used are limited and do not provide sufficient insight.

Generally, the manuscript is well written, the results are well presented and discussed, and the topic of this article is one of high interest currently, namely the investigation of drug resistance biomarkers. However, the findings of the paper need to be supported by further assays, for instance: proteomic analysis, reverse transcription PCR assays, array comparative genomic hybridisation, gene expression analysis.

Further comments and suggestions:

  • The title is rather vague and ambiguous, for instance there is no in-depth discussion in the manuscript about ‘notes for future precision medicine’. In my opinion, the title should specifically describe the main findings of the paper rather than being so broad
  • Line 140 ‘chemotherapy was added in each well’ – chemotherapy should be replaced by chemotherapeutic drug or something similar
  • Materials and Methods Section – a description of the cell treatment with irinotecan has been omitted and it should be added
  • Figure 2 – the percentages of positive cells, commas should be replaced by periods (for instance 0,47 should be 0.47)
  • Discussion section – Line 302 - the sentence ‘So, both DLD-1’s classification as an MSI cell line and its resistance to docetaxel, showed in Figure 4, could justify the high expression of this protein in the DLD-1 cell line’ should be rephrased, as Figure 4 depicts cellular resistance to oxaliplatin; also, the reference to microsatellite instability in relation to docetaxel treatment seems irrelevant in this context

Author Response

Contact Name: Carmen Ramírez Castillejo
Title: Doctor in Cell Biology

Address: Cancer Stem Cell laboratory, HST Research Group.

Centro de Tecnología Biomédica

Dpto. of Biotechnology-VB. ETSIAAB.

Universidad Politécnica de Madrid (UPM)

Avda/ Puerta de Hierro 2-4. 28040 Madrid. Spain

Dear Reviewer,

First of all, I would like to thank you for your time and effort reviewing the manuscript. All of your comments were of great interest, allowing us to improve different aspects of the research article. In this letter we detail the main changes to the article in answer to your comments. We have tried to incorporate as many changes as possible attending to your suggestions, and we hope it is now more suitable for publication in this Special Issue from Biomedicines.

POINT BY POINT CHANGES:

The manuscript by Sánchez-Díez et al. assesses the relation between the expression of ten cancer stem cell biomarkers and drug resistance to oxaliplatin in three colorectal cancer cell lines. The choice of model cancer cell lines is reasonable in that they have different metastatic potential and the expression of the biomarkers is sufficiently diverse to allow for relevant conclusions, yet the methods used are limited and do not provide sufficient insight.

Generally, the manuscript is well written, the results are well presented and discussed, and the topic of this article is one of high interest currently, namely the investigation of drug resistance biomarkers. However, the findings of the paper need to be supported by further assays, for instance: proteomic analysis, reverse transcription PCR assays, array comparative genomic hybridisation, gene expression analysis.

Thank you very much for your comments, we have performed RT-qPCR assays for the biomarkers that presented changes between cell lines in the previous flow cytometry experiments. As expected, both techniques show similar results, which have been incorporated into a new figure 3. As you may know, it was difficult for us to obtain these results in the 10 day revision period, but the effort has been worth it.

Further comments and suggestions:

  • The title is rather vague and ambiguous, for instance there is no in-depth discussion in the manuscript about ‘notes for future precision medicine’. In my opinion, the title should specifically describe the main findings of the paper rather than being so broad

We have changed the title to something more specific to the content of the article: “Implication of different tumor biomarkers in drug resistance and invasiveness in primary and metastatic colorectal cancer cell lines.”

  • Line 140 ‘chemotherapy was added in each well’ – chemotherapy should be replaced by chemotherapeutic drug or something similar

You are right, this has been corrected.

  • Materials and Methods Section – a description of the cell treatment with irinotecan has been omitted and it should be added

The irinotecan concentration employed was described in Section 2.4 (now 2.6). Cells were not pre-treated with irinotecan. The experiment was the following: cells were exposed to oxaliplatin and evaluated by MTT to determine whether the cell lines were equally or more resistant to oxaliplatin and irinotecan. The purpose of this last experiment was to confirm that exposure to oxaliplatin does not affect irinotecan efficacy.

This experiment was explained in lines 314-324. We have added “oxaliplatin” when referring to previously treated cells to clarify that they are only pre-treated with oxaliplatin. Please let us know if further explanation is needed.  

  • Figure 2 – the percentages of positive cells, commas should be replaced by periods (for instance 0,47 should be 0.47)

You are right, this has been modified.

  • Discussion section – Line 302 - the sentence ‘So, both DLD-1’s classification as an MSI cell line and its resistance to docetaxel, showed in Figure 4, could justify the high expression of this protein in the DLD-1 cell line’ should be rephrased, as Figure 4 depicts cellular resistance to oxaliplatin; also, the reference to microsatellite instability in relation to docetaxel treatment seems irrelevant in this context

We have considered your comment and modified it in the corresponding section. We wanted to show that some other authors have related MSI with resistance to other drugs, which correlates with our results, but agree it might have simply led to confusion.  

Finally, we wanted to mention that a new figure has been added (Figure 8) to evaluate the role of Trop2 in drug resistance. The other reviewer pointed out this was a promising result due to the different expression between the SW cell lines and DLD1 cell line, with expression being higher in the drug resistant cell line. We considered this point really interesting and performed these assays to incorporate them in the article. The article’s English has also been revised (please see the corresponding Certificate of Proofreading) as was suggested.

We hope these corrections are satisfactory for you and look forward to your response. Thank you for your time.

Best Regards,

Carmen Ramírez Castillejo

Round 2

Reviewer 1 Report

Marta Sánchez-Díez et al. answered all my questions and considered my suggestions.

Manuscript could be accepted in the current form although please correct one sentence in line 246. („Finally, the TACSTD2 expression profile was the same as in the cytometry assay, with no expression in the SW-480 and SW-620 cell lines and significantly different expression in DLD-1...”). Do not use „was the same” because in case of the DLD-1 cells it is not exactly the same values, use the word „similar”.

Reviewer 2 Report

The authors revised their manuscript to comply with my comments and suggestions. Therefore, I recommend publication of the manuscript.